# *Citrus medica* and *Cinnamomum zeylanicum* Essential Oils as Potential Biopreservatives against Spoilage in Low Alcohol Wine Products

**DOI:** 10.3390/foods9050577

**Published:** 2020-05-04

**Authors:** Gregoria Mitropoulou, Anastasios Nikolaou, Valentini Santarmaki, Georgios Sgouros, Yiannis Kourkoutas

**Affiliations:** Laboratory of Applied Microbiology and Biotechnology, Department of Molecular Biology and Genetics, Democritus University of Thrace, GR-68100 Alexandroupolis, Greece; gmitropo@mbg.duth.gr (G.M.); anikol@mbg.duth.gr (A.N.); vsantar@mbg.duth.gr (V.S.); gsgouros@mbg.duth.gr (G.S.)

**Keywords:** biopreservatives, essential oils, *Citrus medica*, *Cinnamomum zeylanicum*, wine spoilage, low alcohol wines

## Abstract

Low alcohol wine is a new entry in the global wine market, due to the increase in consumers’ concern for health, economic and modern lifestyle issues. As low alcohol products are prone to spoilage, the adoption of natural-derived products with antimicrobial activity as biopreservatives seems to be an intriguing alternative. Thus, the aim of the present study was to investigate the possible antimicrobial properties of *Citrus medica* and *Cinnamomum zeylanicum* essential oils (EOs) and assess their commercial prospective in the wine industry. The main constituents identified by GC/MS analysis were limonene (38.46%) and linalool (35.44%) in *C. medica* EO, whereas trans-cinnamic-aldehyde (63.58%) was the dominant compound in *C. zeylanicum* EO. The minimum inhibitory (MIC), non-inhibitory (NIC) and minimum lethal concentration (MLC) values against common wine spoilage microbes were initially determined. Subsequently, their efficiency was further validated in low alcohol (~6% vol) wines, either separately or in combination at 0.010% (*v*/*v*), as well as in wines deliberately inoculated with *Gluconobacter cerinus*, *Oenococcus oeni*, *Pediococcus pentosaceus*, *Dekkera bruxellensis*, *Candida zemplinina*, *Hanseniaspora uvarum*, *Pichia guilliermondii* or *Zygosaccharomyces bailii*. EO addition led to considerable spoilage and microbial growth delay during storage at room or refrigerated temperature, suggesting their potential use as wine biopreservatives.

## 1. Introduction

Low alcohol wines constitute a new, fast-growing commercial area, due to major public concerns about the serious long-term health effects associated with alcohol consumption, together with consumers’ preferences, which are mostly directed by modern lifestyle and economic reasons. However, low alcohol products are sensitive to spoilage and contain considerable amounts of chemical preservatives with well-known toxicity, such as sulfur dioxide (E220) and potassium metabisulfite (E224) [1,2]. Taking into account the daily wine consumption in Western countries, it is reasonable to consider that the average consumer is often subjected to levels well over the amount generally regarded as safe [3]. Thus, the use of natural agents with antimicrobial activity as biopreservatives, such as essential oils (EOs), might be of interest [4,5,6].

*Citrus medica* (citrus) and *Cinnamomum zeylanicum* (cinnamon) are plants with a great potential in the food industry, as several bioactive compounds with considerable beneficial effects are present in different plant parts [7,8]. Both citron and cinnamon and their derivatives are used in carbonated drinks and alcoholic beverages as raw materials and/or flavoring agents. Their isolated EOs were previously tested for potential antimicrobial activity [9,10,11,12,13,14] and validated in food systems [8,15]. However, their application as biopreservatives in wines is lacking.

An appropriately designed approach for the incorporation of EOs into foods (formulation strategies, efficiency and sensory issues) is a key factor for the development of novel products. Their insertion into a food matrix is considered an additional inherent factor for the delay of spoilage onset. Although several attempts have been focused on the successful addition of EOs and plant extracts on various foods [16,17], only a very limited number of products are available on the market, mainly due to incompatibility with food taste, ineffectiveness owing to the interaction of bioactive substances with food components, their intense aroma [18], etc.

Hence, the aim of the present study was to assess the industrial potential of *Citrus medica* and *Cinnamomum zeylanicum* EOs to be used in low alcohol wines as a natural antimicrobial agent against common spoilage bacteria and yeasts/molds. Data suggesting significant extension of the product’s shelf-life and repression of microbial growth after deliberate inoculation are presented.

## 2. Materials and Methods

### 2.1. Extraction of EOs

*Citrus medica* (citrus) fruits were harvested during October-November 2017. EO was obtained by hydrostillation using a Dean Stark apparatus, in which 12.06 kg of peeled citrus fruits (peeled and chopped into small pieces) were placed along with 6 L of distilled water (VIORYL S.A. Chemical and Agricultural Industry, Scientific Research S.A., Afidnes, Greece) After distillation (8 h, 90–100 °C), 7 g of *Citrus medica* EO were collected.

*Cinnamomum zeylanicum* (cinnamon) EO was supplied by Charabot S.A. (Grasse, France).

### 2.2. Microbial Strains and Culture Media

Oenococcus oeni commercial starter, Pediococccus pentosaceus G22NM13, Gluconobacter cerinus A1M6, Dekkera bruxellensis γΜC2.7, Candida zemplinina E228NL1, Hanseniaspora uvarum E15PL39, Pichia guilliermondii A10W20 and Zygosaccharomyces bailii BgW2 (Y-4) (kindly provided by Dr. Nisiotou A., Athens Wine Institute, ELGO-DIMITRA, Greece) were used in the present study.

*O. oeni* and *P. pentosaceus* were grown in MRS broth (LabM, UK) at 30 °C for 24 h, under anaerobic conditions (Anaerobic Jar 2.5 L, Merck Millipore, USA AnaeroGen 2.5 L Sachets, Oxoid, UK).

*G. cerinus* was grown in suitable culture broth (100 g/L glucose, 20 g/L yeast extract, 20 g/L CaCO_3_) at 30 °C for 24 h.

*D. bruxellensis*, *C. zemplinina*, *H. uvarum*, *P. guilliermondii* and *Z. bailii* were grown in YPD broth (yeast extract 10 g/L, peptone 20 g/L and dextrose 20 g/L) at 30 °C for 24–48 h.

### 2.3. Low Alcohol Wine Production

Low alcohol wine (~6% vol) was produced by free kefir culture using concentrated must of “Muscat Hamburg” grape variety supplied by Tyrnavos Cooperative Winery and Distillery (Tyrnavos, Greece), as recently described [19].

### 2.4. Wine Supplementation with EOs

The produced wines were supplemented with the EOs either separately (0.010% of each oil; 85 or 99 ppm for *Citrus medica* or *C. zeylanicum* EOs, respectively) or in combination (0.010% oil mixture consisting of equal volumes from each EO resulting in 92 ppm of EO mixture) and transferred to sterilized containers. 

After the addition of the EOs, the wines were evaluated for aroma, taste and overall quality attributes using locally approved protocols in our laboratory, as previously reported [19]. All samples were served at 12–15 °C and a blind test in a colored glass under low light was followed.

Then, (a) microbial spoilage, and (b) microbial growth after deliberate inoculation with spoilage microorganisms, were monitored at room (18–20 °C) or low temperature (4 °C), as described below.

### 2.5. Analytical Procedures

#### 2.5.1. GC/MS Analysis

GC/MS analysis was carried out in a GC-MS (GC: 6890A, Agilent Technologies, USA; MSD: 5973, Agilent Technologies) using a Factor Four VF 1ms column (25 m, 0.2 mm i.d., 0.33 μm film thickness, Agilent Technologies), as described previously [10]. Identification was based on the comparison of the retention times and mass spectra of the volatile compounds to Willey/NIST 0.5 and in-house created libraries, as well as on the determination of kovats’ retention indexes (KI) and comparison with those available in the literature.

#### 2.5.2. Antimicrobial Assays

##### Screening of EOs Antimicrobial Activity by the Disc Diffusion Assay

The antimicrobial activity of *C. medica* and *C. zeylanicum* EOs, either separately or in combination (EO mixture), was initially tested using the disk diffusion assay, as described previously [10]. In brief, 0.1 mL of a microbial suspension in ¼ strength Ringer’s solution (LabM) was spread on the suitable agar culture medium, providing initial inoculums of 10^5^ or 10^7^ cfu/mL. Then, sterile paper disks (Whatman no. 2) of 5 mm diameter were placed onto the inoculated agar surface containing 5 μL of the EOs or the EO mixture. After incubation, the inhibition zones were measured in mm. Erythromycin was used as positive control for *O. oeni* and *P. pentosaceus*, gentamycin for *G. cerinus* and voriconazole for yeasts. All experiments were carried out in four replicates.

##### Determination of Minimum Inhibitory (MIC), Non-Inhibitory (NIC) and Minimum Lethal (MLC) Concentrations

Minimum inhibitory (MIC) and non-inhibitory concentration (NIC) for *O. oeni* and *P. pentosaceus* were determined as previously described [10]. Broth media with no inoculum and inoculated broths with no EOs were used as negative controls [10]. The calculation of MIC and NIC values was based on the Lambert–Pearson model (LPM) [20,21].

The common microdilution method [22,23,24] was followed for screening the activity of the EOs against *G. cerinus*, *D. bruxellensis*, *C. zemplinina*, *H. uvarum*, *P. guilliermondii* and *Z. bailii* and MIC determination (NIC determination was impossible), as application of the LPM model was not possible due to the high turbidity of *G. cerinus* culture broth, known as the “milky” effect [25] and yeast cell sedimentation. After incubation, cell growth was observed at 610 nm. Erythromycin was used as positive control for *O. oeni* and *P. pentosaceus*, gentamycin for *G. cerinus* and voriconazole for yeasts.

Minimum lethal concentration (MLC) was determined as described before [10].

All experiments were carried out in four replicates.

##### Antimicrobial Activity in Wines-Containing EOs

Spoilage of low alcohol wines supplemented with EOs and stored at room temperature (18–20 °C) or at 4 °C was monitored by determining bacteria and yeast/molds counts on BHI Agar (LabM) at 37 °C and Malt extract agar (LabM) at 30 °C, respectively, for 2–5 days.

Similarly, low alcohol wines containing either *C. medica* or *C. zeylanicum* EO or in combination (EO mixture) were deliberately spiked with *O. oeni*, *P. pentosaceus*, *G. cerinus*, *D. bruxellensis*, *C. zemplinina*, *H. uvarum*, *P. guilliermondii* or *Z. bailii*, separately (initial inoculum of 10^3^ cfu/mL was applied) and stored at room temperature (18-20 °C) and at 4 °C. Samples were collected at various intervals to monitor the levels of the inoculated strains.

In both experiments, wines with no EOs were used as controls.

All experiments were carried out in triplicate.

### 2.6. Statistical Analysis

The mean values are presented and standard deviation in MIC and NIC values determined by the Lambert–Pearson model (LPM) was calculated by Figure P.2.1 software (Figure P Software Incorporated, Hamilton, ON, Canada). Standard deviation estimation was not applicable when the common microdilution method was used.

The results were analyzed for statistical significance with analysis of variance (ANOVA). Duncan’s multiple range test was applied to determine significant differences (coefficients, ANOVA tables and significance (*p* < 0.05) were computed using Statistica v.10.0, StatSoft, Tulsa, OK, USA).

## 3. Results and Discussion

### 3.1. GC/MS Analysis

GC/MS analysis provided data about the percentage content of the volatile compounds in the EOs and not their actual concentration. The main compounds detected by GC/MS were limonene (38.46%) and linalool (35.44%) in *Citrus medica* EO, whereas *trans*-cinnamic-aldehyde accounted for 63.58% of the area in *Cinnamomum zeylanicum* EO (Table 1).

### 3.2. Antimicrobial Assays

Considering that composition and thus effectiveness of EOs depends greatly upon a number of cultivation, environmental and climate factors [26,27,28], the antimicrobial activity of the two EOs separately or their mixture against *Oenococcus oeni*, *Pediococcus pentosaceus*, *Gluconobacter cerinus*, *Dekkera bruxellensis*, *Candida zemplinina*, *Hanseniaspora uvarum*, *Pichia guilliermondii* and *Zygosaccharomyces bailii* was initially confirmed using the disk diffusion method (data not shown). Subsequently, MIC and NIC (when applicable) values against the microbial species usually responsible for wine spoilage were assessed, as their precise determination is crucial for the food industry, in order to regulate the optimum amount of the antimicrobial agent to prevent microbial spoilage. The effective growth inhibition of both oils, separately or in combination, against all microbial species (Table 2) was documented, although MIC, NIC (when applicable) and MLC values were significantly (*p* < 0.05) higher compared to erythromycin, gentamycin and voriconazole (Table 2), which were used as positive controls.

Similarly, growth inhibition of *C. medica* EO against *Aspergillus niger*, *Aspergillus flavus*, *Saccharomyces cerevisiae*, *Candida albicans* and *Pediococcus dextrinicus* [10,29] and *C. zeylanicum* EO against *Candida albicans* and *Pediococcus dextrinicus* has been also evidenced previously [22,29].

### 3.3. Sensory Trials

To determine the maximum concentration of EOs resulting in products with acceptable sensory characteristics, an evaluation test was performed. According to the results (data not shown), the addition of EOs significantly affected (*p* < 0.05) all quality attributes.

Wines supplemented with 0.010% *C. medica* EO were characterized by a fruity and citrus-like aroma, while increasing concentrations of the added oil resulted in a pungent aroma and repellent taste. A sweet or sweet/sour taste was mostly predominant when *C. zeylanicum* was added (at 0.010% vol) and a warming effect was noticed when its concentration was increased. A mixture of both EOs at equal concentrations (0.005%) resulted in a pleasant fruity/wine-like aroma and sweet/sour taste.

In all cases, concentrations higher than 0.010% were rejected, as they masked the wine flavor and oil residues formed an obvious layer on the top of the wine.

### 3.4. Antimicrobial Activity of C. medica and C. zeylanicum EOs in Low Alcohol Wines

Primarily, the effect of *C. medica* and *C. zeylanicum* EOs on the extension of the preservation time was investigated both separately and in combination. The oils were introduced in low alcohol wines (~6% vol) in concentrations (0.010 % *v*/*v*) lower than the MIC, as higher levels resulted in sensory faults. The results are presented in Figure 1. In both temperatures, a significant increase of the shelf-life was noticed. Specifically, at room temperature, spoilage in wines containing *C. medica* or *C. zeylanicum* EO or their mixture was observed after 18, 74 or 67 days, respectively, in contrast to 9 days in the control (wine with no EO). Likewise, at 4 °C, the corresponding preservation days for wines supplemented with *C. medica* or *C. zeylanicum* EO or their mixture were 74, 109 and 88 days, while spoilage in the control sample was evidenced after 39 days.

Subsequently, the resistance of low alcohol wines supplemented with *C. medica* and *C. zeylanicum* EOs against the growth of spoilage microbes was further investigated. In this vein, wines supplemented with *C. medica* or *C. zeylanicum* EOs, either separately or in combination and wines with no EOs were deliberately inoculated with *O. oeni*, *P. pentosaceus*, *G. cerinus*, *D. bruxellensis*, *C. zemplinina*, *H. uvarum*, *P. guilliermondii* and *Z. bailii*, separately. Microbial growth was monitored at room temperature or refrigerated temperature (Figure 1 and Figure 2). The results showed that in all cases microbial counts were significantly (*p* < 0.05) lower in wines supplemented with the EOs.

Cinnamon and citrus EOs or powders were previously proposed as preventing agents against microorganism-induced food spoilage and to secure microbial safety in fruit juices [30,31]. Likewise, the addition of cinnamon powder in unpasteurized apple cider resulted in 2 logcfu/mL reduction of *Escherichia coli* O157:H7 [32]. Additionally, lemongrass, cinnamon, or geraniol EOs inactivated *Salmonella enteritidis*, *E. coli*, and *Listeria innocua* in unpasteurized apple and pear juices [33]. Similarly, recent reports suggested that citrus EOs may represent a promising alternative for pathogens in fruit juices [34]. Specifically, *Citrus limon* oil was effective against *E. coli* O157:H7 and *S.** enterica* in apple juice [35], while it led to germination inhibition and outgrowth of *Acinetobacter acidoterrestris* spores in lemon juice under refrigerated storage [36]. In the same manner, cinnamon EO in tyndallized carrot broth inhibited the spore germination of psychrotrophic *Bacillus cereus* at 8 or 12 °C, but not at 16 °C [37]. Other studies demonstrated the effective growth inhibition of *E. coli* O157:H7 and *S. enterica* in wine marinades supplemented with plant extracts and EOs from oregano, thyme and its derivatives and were considered suitable for meat products [38,39].

Hence, to the best of the authors’ knowledge, the present study reports for the first time the effectiveness of *C. medica* and *C. zeylanicum* EOs in suppressing microbial growth in deliberately spiked low alcohol wines.

## 4. Conclusions

The results showed that wine supplementation with *C. medica* or *C. zeylanicum* EOs separately or in combination led to considerable spoilage and microbial growth delay, promoting their use as effective antimicrobial agents in the wine industry. The observed effects may be attributed to the main bioactive constituents of the oils, but further testing using pure standard compounds may be useful to validate such a hypothesis, as well as to identify potential synergistic or antagonistic interactions, leading to even more specific industrial applications.

## Figures and Tables

**Figure 1 foods-09-00577-f001:**
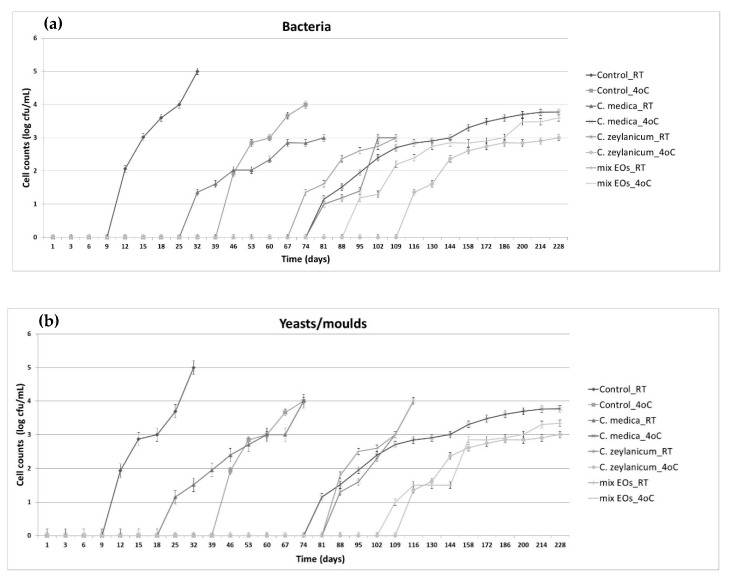
Effect of *Citrus medica* and *Cinnamomum zeylanicum* EOs on spoilage of low alcohol wines (~6 % vol) during storage at room temperature (18–20 °C) or at 4 °C. Wines were supplemented with either *C. medica* or *C. zeylanicum* EO (0.010%) or their mixture (0.005% of each oil). Wine with no EOs was used as control. (**a**) Bacteria counts, (**b**) Counts of yeasts/molds. Counts below the detection limit (<10 cells/mL) are indicated with 0 value.

**Figure 2 foods-09-00577-f002:**
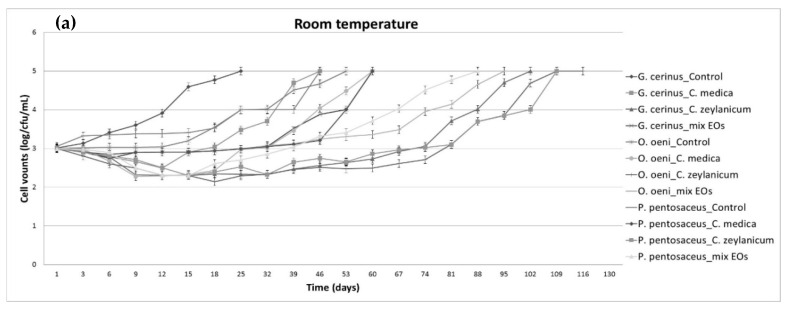
Effect of *Citrus medica* and *Cinnamomum zeylanicum* EOs on microbial growth in deliberately spiked low alcohol wines (~6 % vol) during storage at room temperature (18–20 °C) and at 4 °C. Wines were supplemented with either *C. medica* or *C. zeylanicum* EO (0.010%) or their mixture (0.005% of each oil). Wine with no EO was used as control. *Gluconobacter cerinus*, *Oenococcus oeni* and *Pediococcus pentosaceus* counts at room temperature (**a**) or at 4 °C (**b**). *Dekkera bruxellensis*, *Candida zemplinina*, *Hanseniaspora uvarum*, *Pichia guillermndii* and *Zygosaccharomyces bailii* counts at room temperature (**c**) or at 4 °C (**d**).

**Table 1 foods-09-00577-t001:** Relative percentage (%) area of volatiles identified in *Citrus medica* and *Cinnamomum zeylanicum* essential oils (Eos) by GC/MS analysis.

Compounds	KRI ^1^	*C. medica* (% area)	*C. zeylanicum* (% area) ^2^
3-Methyl-2-buten-1-ol	740	Tr	-
*cis*-3-Hexenol	831	Tr	-
Hexanol	843	Tr	-
Isoamyl acetate	854	Tr	-
Styrene	870	-	Tr
Thujene	920	0.29	0.30
*α*-Pinene	928	0.81	1.56
Camphene	940	-	0.51
Hexyl acetate	954	Tr	-
6-Methyl-5-hepten2-one	958	0.12	-
Sabinene	962	Tr	Tr
*β*-Pinene	967	0.79	0.47
Octanal	977	-	Tr
Myrcene	979	1.23	-
*α*-Phellandrene	996	Tr	1.36
*d*-3-Carene	1004	-	0.13
*a*-Terpinene	1008	0.16	-
*p*-Cymene	1010	0.64	3.15
*β*-Phellandrene	1017	-	3.67
Limonene	1020	38.46	-
*cis*-Ocimene	1022	Tr	0.07
Acetophenone	1028	-	Tr
*trans*-Ocimene	1032	0.19	-
*γ*-Terpinene	1043	7.96	Tr
Sabinene hydrate	1046	-	Tr
Epoxy linaloοl	1052	0.10	Tr
Dehydro-*p*-cymene	1066	-	Tr
Terpinolene	1073	0.46	0.14
Nonanal	1077	Tr	-
Linaloοl	1084	35.44	4.22
Isoamyl valerate	1086	-	Tr
Menth-2-en-1-ol	1103	-	Tr
iso-Nerol	1104	-	Tr
Camphor	1114	-	Tr
iso-Geraniol	1118	Tr	-
Phenyl propionaldehyde	1119	-	0.28
*β*-Terpineol	1121	Tr	-
Citronellal	1127	Tr	-
iso-Citral	1136	Tr	-
Menthadienol	1139	Tr	-
Borneol	1141	-	Tr
Cryptone	1147	-	Tr
4-Terpineol	1153	0.54	0.28
*α*-Terpineol	1164	2.09	0.50
Hexyl butyrate	1171	Tr	-
*cis*-Cinnamic aldehyde	1174	-	0.48
Decanal	1180	-	Tr
3-Phenylpropanol	1194	-	Tr
*o*-Anisaldehyde	1199	-	Tr
Nerol	1206	1.58	-
*cis*-Citral	1210	1.88	-
*trans*-Cinnamic-aldehyde	1230	-	63.58
Geraniol	1231	2.34	-
*trans*-Citral	1237	3.09	-
Phellandral	1241	-	Tr
Safrole	1255	-	Tr
Cinnamic alcohol	1263	-	0.13
Thymol	1264	-	Tr
Carvacrol	1273	-	Tr
Eugenol	1322	-	5.05
Terpinyl acetate	1325	Tr	-
Hydrocinnamyl acetate	1332	-	Tr
Neryl acetate	1339	0.27	-
*α*-Cubebene	1339	-	Tr
*cis*-Cinnamyl acetate	1350	-	Tr
Geranyl acetate	1358	0.52	-
Copaene	1364	-	0.74
Coumarine	1369	-	Tr
*β*-Elemene	1378	-	Tr
*trans*-Cinnamyl acetate	1403	-	2.36
Caryophyllene	1404		5.92
*α*-Humulene	1439	-	1.08
Caryophyllene oxide	1455	-	0.98
*o*-Methoxy cinnamic aldehyde	1476	-	0.40
Valencene	1478	Tr	-
Acetyl eugenol	1483	-	0.25
*β*-Bisabolene	1495	0.10	-
Geranyl-iso-butyrate	1515	Tr	-
Geranyl butyrate	1538	Tr	-
*trans*-Nerolidol	1545	Tr	-
Benzyl benzoate	1713	-	1.54

1: Kováts Retention Indices, 2: Τr: traces (<0.10%).

**Table 2 foods-09-00577-t002:** Minimum inhibitory (MIC), non-inhibitory (NIC) and minimum lethal concentration (MLC) (mg/L) of *Citrus medica* and *Cinnamomum zeylanicum* EOs and their mixtures against common wine spoilage microbes. Erythromycin was used as positive control for *Oenococcus oeni* and *Pediococcus pentosaceus*, gentamycin for *Gluconobacter cerinus* and voriconazole for yeasts (standard deviation ranged in zero values if not shown). Application of the Lambert–Pearson model (LPM) and thus NIC determination for *Gluconobacter cerinus*, *Dekkera bruxellensis*, *Candida zemplinina*, *Hanseniaspora uvarum*, *Pichia guilliermondii* and *Zygosaccharomyces bailii* was impossible, due to the high turbidity of *G. cerinus* culture broth (“milky” effect) and yeast cell sedimentation.

Microbial Species	*C. medica* EO			*C. zeylanicum* EO			EOs Mixture			Erythromycin			Voriconazole			Gentamycin		
	MIC	NIC	MLC	MIC	NIC	MLC	MIC	NIC	MLC	MIC	NIC	MLC	MIC	NIC	MLC	MIC	NIC	MLC
*G. cerinus*	2544	-	10176	1245	-	5978	2305	-	9222	-	-	-	-	-	-	4	-	8
*O. oeni*	2097 ± 25	1081 ± 34	10176	1037 ± 30	202 ± 40	5978	2041 ± 27	687 ± 18	9222	0.24 ± 0.001	0.06 ± 0.001	1	-	-	-	-	-	-
*P. pentosaceus*	2506 ± 42	1530 ± 25	10176	1549 ± 40	361 ± 50	5978	2089 ± 18	234 ± 9	9222	0.12 ± 0.001	0.06 ± 0.002	0.5	-	-	-	-	-	-
*D. bruxellensis*	4240	-	12720	125	-	747	2305	-	13832	-	-	-	1	-	4	-	-	-
*C. zemplinina*	636	-	1696	374	-	1494	350	-	1844	-	-	-	1	-	4	-	-	-
*H. uvarum*	530	-	1696	498	-	1993	350	-	1844	-	-	-	1	-	4	-	-	-
*P. guilliermondii*	636	-	1696	374	-	2491	350	-	1844	-	-	-	1	-	4	-	-	-
*Z. bailii*	530	-	1272	498	-	5978	1752	-	7377	-	-	-	1	-	4	-	-	-

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
