# Peer review of "Citrus medica and Cinnamomum zeylanicum Essential Oils as Potential Biopreservatives against Spoilage in Low Alcohol Wine Products"

_foods, 2020, doi:10.3390/foods9050577_

Round 1

Reviewer 1 Report

The manuscript "Citrus medica and Cinnamomum zeylanicum Essential  Oils as Potential Biopreservatives Against Spoilage
 in Low Alcohol Wine Products" describes the study performed by the authors on Citrus medica and Cinnamomum zeylanicum two Essential oils, characterized by GC MS and for whose the antimicrobial activity and potential use was assessed in the wine industry. The oils were tested singularly and in mix against different common wine spoilage microbes. Then, their efficiency 
 was further validated in low alcohol (~6% vol) wines, either separately or in combination at 0.010% (v/v) deliberately inoculated with Oenococcus oeni, Pediococcus pentosaceus, Gluconobacter  cerinus, Dekkera bruxellensis, Candida zemplinina, Hanseniaspora uvarum, Pichia guilliermondii , Zygosaccharomyces bailii. The activity of the EOs caused a significant delay as regard as spoilage and microbial growth. the test were performed both at room and refrigerated temperatures. the work on the Whole is interesting.

Some suggestion: please, check all around the text the names of plant and microbial species you used. Sometimes they were not written in italics.

line 102, please, provide details about the methods you used for the antimicrobial tests. The reader shouldn't be obliged to find the reference to reply your method.

line 143 I'ld add near "the antimicrobial activity " : "of the two EOs"

Author Response

  1. Comment: Some suggestion: please, check all around the text the names of plant and microbial species you used. Sometimes they were not written in italics.

Answer: Following the reviewer’s comment, the text has been checked and plant and microbial species is written in italics in the revised manuscript.

  1. Comment: line 102, please, provide details about the methods you used for the antimicrobial tests. The reader shouldn't be obliged to find the reference to reply your method.

Answer: Details about the method followed is provided in the revised manuscript (l. 103-108), as suggested by the reviewer.

  1. Comment: line 143 I'ld add near "the antimicrobial activity " : "of the two EOs"

Answer: The title has been modified according to the reviewer suggestion (l. 154).

Reviewer 2 Report

The paper »Citrus medica and Cinnamomum zeylanicum Essential Oils as Potential Biopreservatives Against Spoilage in Low Alcohol Wine Products” by  Gregoria Mitropoulou et al. reports interesting new findings on the possibility of preventing microbial low alcohol wine spoilage by adding well specified quantities of Citrus medica and Cinnamomum zeylanicum derived essential oils.  The results clearly indicate that addition of low concentrations of these EOs can considerably prolong the shelf-life of such beverages because the growth of various microbes involved in the spoilage can be significantly impaired.

Very importantly from a practical point of view, the authors also included tests on the alterations of smells and tastes of the beverages when adding EOs. In this way, they open the way to potential industrial uses of these EOs in combination with selected beverages. It turns out that the use of EOs in such beverages needs very carefully tuning to match future users’ tastes.

The paper is well written and well presented. The results obtained are convincing. However, a few points (see general specific comments below) and a number of specific comments (see below) need to be considered .

General comments

  1. The references cited (7-8) are very partially matching the claim that these publications prove that “their antimicrobial activities” were tested. The authors should consider the following references that appear more suitable. See for example,

For Citrus: i ZH, Cai M, Liu YS, Sun PL, Luo SL. Antibacterial Activity and Mechanisms of Essential Oil from Citrus medica L. var. sarcodactylis. Molecules. 2019 Apr 22;24(8). pii: E1577. doi: 10.3390/molecules24081577.

             For Cinnamon: Qing Liu, Xiao Meng, Ya Li, Cai-Ning Zhao, Guo-Yi Tang, and Hua-Bin Li.

             Antibacterial  and antifungal activities of Spices. Int J Mol Sci. 2017 Jun; 18(6): 1283.

             doi: 10.3390/ijms18061283 ; Vasconcelos NG, Croda J, Simionatto S. Antibacterial mechanisms of cinnamon and its constituents: A review. Microb Pathog. 2018 Jul;120:198-203. Doi : doi:10.1016/j.micpath.2018.04.036.;

             Kumar S, Kumari R, Mishra S. J Pharm Pharmacol. Pharmacological properties and their medicinal uses of Cinnamomum: a review.  2019 Dec;71(12):1735-1761. doi: 10.1111/jphp.13173.

  1. The statement Lines 41-42 may not be quite right: see Chhikara, N.; Kour, R.; Jaglan, S.; Gupta, P.; Gat, Y.; Panghal, A. Citrus medica: Nutritional, phytochemical composition and health benefits—A review. Food Funct. 2018, 9, 1978–1992.

              Apparently, “Citrus medica is an underutilized fruit plant”. Main bioactive

             compounds include limonene and linalool among many others. It is used already in

             some carbonated drinks and alcoholic beverages.

  1. The authors (and the editor) should decide whether they use throughout the paper italics for Latin names of all microbes (see Lines 63-65) throughout the paper.
  2. Line 124: Most experiments were performed in four replicates. In seems to be wise to clearly indicate those that were done in triplicate only.
  3. Table 1: the Heading media and C. zeylanicum looks somewhat aukward and could be misleading. (better: outspelled correctly!)
  4. In Materials and Methods the positive controls (Line 151) were not mentioned.
  5. In the paragraph of3.2 the microbes should be fully spelled out.
  6. The authors might want to indicate the range of shelf-life extension in days (without microbial spoilage) gained to substantiate their findings.
  7. The authors seem to assume that the observed beneficial effects of EOs addition may be due to their main constituents. Further testing may be useful to identify whether such a supposition is valid. The results may guide even more specific industrial applications.

Specific comments

Line11: Abstract: Low

Line 15: Citrus medica and Cinnamomum zeylanicum

Line 17: C. medica

Line18: C. zeylanicum

Lines 22-24: Oenococcus oeni, Pediococcus pentosaceus, Gluconobacter

cerinus, Dekkera bruxellensis, Candida zemplinina, Hanseniaspora uvarum, Pichia guilliermondii

 or Zygosaccharomyces bailii.

Line 34: sulfur dioxide

Line 35: potassium metabisulfite

Line 35: Taking into account the daily wine (change font size)

36 consumption in Western countries

Line 37: as biopreservatives might be of interest.

Line 47: on the market

Line 53: self-life? (= shelf-life)

Lines 135-137: identical with abstract. Is this intentionally done?

Line 154: Candida albicans

Lines 276-277 do not need to be underlined!

Author Response

General comments

  1. Comment: The references cited (7-8) are very partially matching the claim that these publications prove that “their antimicrobial activities” were tested. The authors should consider the following references that appear more suitable. See for example,

For Citrus: i ZH, Cai M, Liu YS, Sun PL, Luo SL. Antibacterial Activity and Mechanisms of Essential Oil from Citrus medica L. var. sarcodactylis. Molecules. 2019 Apr 22;24(8). pii: E1577. doi: 10.3390/molecules24081577.

For Cinnamon: Qing Liu, Xiao Meng, Ya Li, Cai-Ning Zhao, Guo-Yi Tang, and Hua-Bin Li. Antibacterial  and antifungal activities of Spices. Int J Mol Sci. 2017 Jun; 18(6): 1283.

doi: 10.3390/ijms18061283 ; Vasconcelos NG, Croda J, Simionatto S. Antibacterial mechanisms of cinnamon and its constituents: A review. Microb Pathog. 2018 Jul;120:198-203. Doi : doi:10.1016/j.micpath.2018.04.036.;

Kumar S, Kumari R, Mishra S. J Pharm Pharmacol. Pharmacological properties and their medicinal uses of Cinnamomum: a review.  2019 Dec;71(12):1735-1761. doi: 10.1111/jphp.13173.

Answer:  Following the reviewer’s comment, the suggested references have been included in the revised manuscript (l. 44). 

  1. Comment: The statement Lines 41-42 may not be quite right: see Chhikara, N.; Kour, R.; Jaglan, S.; Gupta, P.; Gat, Y.; Panghal, A. Citrus medica: Nutritional, phytochemical composition and health benefits—A review. Food Funct. 2018, 9, 1978–1992.

Apparently, “Citrus medica is an underutilized fruit plant”. Main bioactive  compounds include limonene and linalool among many others. It is used already in some carbonated drinks and alcoholic beverages.

Answer: Following the reviewer’s comment, the sentence has been rephrased in the revised manuscript (l. 44-45).

  1. Comment: The authors (and the editor) should decide whether they use throughout the paper italics for Latin names of all microbes (see Lines 63-65) throughout the paper.

Answer: The text has been checked and plant and microbial species is written in italics in the revised manuscript, as suggested by the reviewer.

  1. Comment: Line 124: Most experiments were performed in four replicates. In seems to be wise to clearly indicate those that were done in triplicate only.

Answer: Following the reviewer’s suggestion, it is stated which experiments were carried out in triplicate and which in four replicates (l. 108, 109, 124 & 135).

  1. Comment: Table 1: the Heading media and C. zeylanicum looks somewhat aukward and could be misleading. (better: outspelled correctly!)

Answer: Following the reviewer’s comment, the heading of Table 1 has been corrected in the revised manuscript (l. 150-151).

  1. Comment: In Materials and Methods the positive controls (Line 151) were not mentioned.

Answer: Following the reviewer’s comment, the positive controls are mentioned in the Materials and Methods Section in the revised manuscript (l. 107-108 & 121-122).

  1. Comment: In the paragraph of3.2 the microbes should be fully spelled out.

Answer: Following the reviewer’s suggestion, the microbes are fully spelled out In the paragraph 3.2 (l. 156-158 & 166-168).

  1. Comment: The authors might want to indicate the range of shelf-life extension in days (without microbial spoilage) gained to substantiate their findings.

Answer: Following the reviewer’s suggestion, the range of shelf-life extension in days (without microbial spoilage) is reported and highlighted in the revised manuscript (l. 193-198).  

  1. Comment: The authors seem to assume that the observed beneficial effects of EOs addition may be due to their main constituents. Further testing may be useful to identify whether such a supposition is valid. The results may guide even more specific industrial applications.

Answer: We agree with the reviewer’s comment. Therefore, a relevant paragraph has been added in the conclusions (l. 251-254).

Specific comments

  1. Comment: Line11: Abstract: Low.

Answer: Following the reviewer’s comment, the phrase has been corrected in the revised manuscript (l. 11).

  1. Comment: Line 15: Citrus medicaand Cinnamomum zeylanicum.

Answer: Following the reviewer’s comment, Citrus medica and Cinnamomum zeylanicum are written in italics in the revised manuscript (l. 15).

  1. Comment: Line 17: C. medica.

Answer: Following the reviewer’s comment, Citrus medica is written in italics in the revised manuscript (l. 17).

  1. Comment: Line18: C. zeylanicum.

Answer: Following the reviewer’s comment, Cinnamomum zeylanicum is written in italics in the revised manuscript (l. 18).

  1. Comment: Lines 22-24: Oenococcus oeniPediococcus pentosaceusGluconobactercerinusDekkera bruxellensisCandida zemplininaHanseniaspora uvarumPichia guilliermondiivor Zygosaccharomyces bailii.

Answer: Following the reviewer’s comment, all microbes are written in italics in the revised manuscript (l. 22-24).

  1. Comment: Line 34: sulfur dioxide.

Answer: Following the reviewer’s comment, sulfur dioxide is correctly written in the revised manuscript (l. 34 & 35).

  1. Comment: Line 35: potassium metabisulfite.

Answer: Following the reviewer’s comment, potassium metabisulfite is correctly written in the revised manuscript (l. 35).

  1. Comment: Line 35: Taking into account the daily wine (change font size) 36 consumption in Western countries.

Answer: Following the reviewer’s comment, font size is corrected in the revised manuscript (l. 35).

  1. Comment: Line 37: as biopreservatives might be of interest.

Answer: Following the reviewer’s comment, the sentence has been modified in the revised manuscript (l. 38 & 39).

  1. Comment: Line 47: on the market.

Answer: Following the reviewer’s comment, the phrase has been corrected in the revised manuscript (l. 50).

  1. Comment: Line 53: self-life? (= shelf-life).

Answer: Following the reviewer’s comment, shelf-life is correctly written in the revised manuscript (l. 56).

  1. Comment: Lines 135-137: identical with abstract. Is this intentionally done?

Answer: Following the reviewer’s comment, the phrase has been modified in the revised manuscript (l. 148-149).

  1. Comment: Line 154: Candida albicans

Answer: Following the reviewer’s comment, Candida albicans is correctly written in the revised manuscript (l. 148-149).

  1. Comment: Lines 276-277 do not need to be underlined!

Answer: Following the reviewer’s comment, lines 276-277 (l. 313-315 in the revised manuscript) are not underlined.